# Contribution of host species and pathogen clade to snake fungal disease hotspots in Europe
Gaëlle Blanvillain [1] ✉, Jeffrey M. Lorch[2], Nicolas Joudrier[3,4,5], Stanislaw Bury[6,7], Thibault Cuenot [8], Michael Franzen[9], Fernando Martínez-Freiría [10,11], Gaëtan Guiller[12], Bálint Halpern[13,14,15], Aleksandra Kolanek[7,16], Katarzyna Kurek[17], Olivier Lourdais[18,19], Alix Michon[8], Radka Musilová[20], Silke Schweiger[21], Barbara Szulc[7,22], Sylvain Ursenbacher[5,23,24], Oleksandr Zinenko [25] & Joseph R. Hoyt [1]

Infectious diseases are influenced by interactions between host and pathogen, and the number of infected hosts is rarely homogenous across the landscape. Areas with elevated pathogen prevalence can maintain a high force of infection and may indicate areas with disease impacts on host populations. However, isolating the ecological processes that result in increases in infection prevalence and intensity remains a challenge. Here we elucidate the contribution of pathogen clade and host species in disease hotspots caused by *Ophidiomyces ophidiicola*, the pathogen responsible for snake fungal disease, in 21 species of snakes infected with multiple pathogen strains across 10 countries in Europe. We found isolated areas of disease hotspots in a landscape where infections were otherwise low. *O. ophidiicola* clade had important effects on transmission, and areas with multiple pathogen clades had higher host infection prevalence. Snake species further influenced infection, with most positive detections coming from species within the *Natrix* genus. Our results suggest that both host and pathogen identity are essential components contributing to increased pathogen prevalence.

Infectious diseases can shape ecological communities by altering host abundance and distributions across the landscape[1–3]. Disease outcomes are determined by host-pathogen interactions, which are multifaceted and can interact with environmental conditions, creating a mosaic of disease hotspots across broad spatial scales[4–7]. Hotspots of high pathogen prevalence may represent potential areas of continued impacts to host populations, serve as a source for pathogen dispersal, and maintain high propagule pressure within host communities[4–9].

Heterogeneity in innate species susceptibility is recognized as a strong force influencing pathogen transmission and disease impacts for multi-host pathogens[1,10–12]. The distribution of highly susceptible species can determine areas of high prevalence if they are critical in pathogen maintenance[13,14]. However, the disproportionate contribution of a particular species may be modified by differences in community structure, environmental conditions among patches, and variation in pathogen virulence[7,15]. Pathogen replication rates can also differ among strains and across the landscape, producing additional variation in disease prevalence[16,17]. Pathogen strains with high growth rates and virulence may be due to multiple factors, including the

introduction of novel strains to new locations or hosts[18,19], ease or independence of transmission from affected hosts[20,21], and the development of novel mutations or adaptations that facilitate the escape from host resistance[22]. Although the interaction between host species and pathogen identity is rarely examined, theory suggests that the presence of host species with low susceptibility could modify the effects of pathogen strains with high replication rates, creating cold spots of transmission across the landscape[23]. Conversely, pathogen strains with lower replication rates in the presence of more susceptible host species could drive hotspots of infection[24–26].

The fungal pathogen *Ophidiomyces ophidiicola*, that causes snake fungal disease (SFD, also called ophidiomycosis), has been documented in over 42 species of wild snakes across three continents[27–34], and is considered a threat to the conservation of snake populations[35,36]. Clinical signs of disease caused by *O. ophidiicola* can range from mild skin lesions, from which snakes can recover, to severe infections that impair movement, disrupt feeding behavior, and can ultimately lead to death[28,36,37]. Snake fungal disease is known to affect a wide range of snake species, irrespective of their

---

ecological traits and phylogenetic relationships[29]. While snake susceptibility is widespread, the manifestation of clinical signs varies spatially and temporally with higher infection rates often observed at spring emergence in temperate climates and in cooler temperatures where snakes do not hibernate[28,38,39]. Although high mortality rates associated with SFD have been documented in some species of North American snakes[40], a recent review found no indication that the prevalence of *O. ophidiicola* or the associated disease increased over the past decade in Canada[27]. Still, SFD poses challenges in understanding its population-level effects due to the secretive nature of snakes and the resulting low encounter rate, and few studies have examined the effects of SFD on snake survival. Snake population declines have not been reported in Europe, where *O. ophidiicola* has most likely coexisted with host species for longer periods of time[41]. However, limited information is available on *O. ophidiicola* infections across Europe, with just a few individual snakes confirmed to be infected with this pathogen from mainland Europe[30,31,42-44].

Little is known of the origin of *O. ophidiicola*, and to date, three distinct clades of *O. ophidiicola* have been described: clade I, which has been found exclusively in wild snakes in Europe; clade II, which has been reported in wild snakes in North America and Taiwan as well as in captive snakes on multiple continents; and clade III, which has only been found in captive snakes[41]. The estimation of the most recent common ancestor between clade I and II (around 2000 years ago), as well as a lack of nonrecombinant intermediates in North America strongly indicate that *O. ophidiicola* was introduced to North America, potentially through multiple introduction events[41]. More recently, genotyping from a limited number of samples indicated the presence of both clades I and II in Switzerland dating back to at least 1959[44]. In addition, slower growth rates have been reported for clade I strains of *O. ophidiicola*, suggesting that they may be less virulent than clade II strains[30]. However, prevalence and disease severity associated with the two strains and how they influence landscape patterns of disease has not been compared.

To investigate the macroecological patterns of SFD across Europe we examined host and pathogen factors that may influence pathogen prevalence and disease severity across the landscape. We first evaluated the presence of elevated pathogen prevalence (i.e. hotspots) across Europe, and examined differences in pathogen prevalence and disease severity among host species. Finally, we used a model incorporating the interaction effects of host species and pathogen clade to explore factors that may contribute to areas with high pathogen prevalence across the landscape.

## Results

We captured 1254 individual snakes from 21 species representing 6 genera (Fig. 1, Supplementary Table 2). A total of 2628 swabs were collected, including 2357 full body skin swabs and 271 lesion swabs. Overall, *O. ophidiicola* prevalence confirmed by qPCR was 8.7% (*n* = 109 positive snakes) and prevalence was highly variable across the landscape. Observed pathogen prevalence based on qPCR results was highest in sites across Switzerland (results presented as: mean (95% confidence interval); 26.7% (21.1, 32.4)), followed by sites in Ukraine (12.7% (4.3, 21.2)) and Germany (12.5% (4.2, 20.8)). Sites in Poland and the Iberian Peninsula (Spain and Portugal) had the lowest prevalence at 2.7% (0.3, 5.1), 0.0%, and 0.0% respectively, despite comparable sample sizes to other locations (Supplementary Table 2, Supplementary Fig. 1). We found no statistical support that *O. ophidiicola* prevalence varied over the sampling period (Supplementary Fig. 2; date coeff: 0.01 ± 0.00 (0.00, 0.02)).

Hotspots were determined as sites for which estimated prevalence was above the estimated mean *O. ophidiicola* prevalence across all sites (4.3%). Sites above this threshold included four sites in Switzerland (results formatted as the posterior mean ± standard deviation (95% credible interval); sw11, 26% ± 15.5% (3.9, 62.2); sw3, 23.1% ± 14.3% (3.3, 57.2); sw5, 20.2% ± 13.0% (2.6, 51.0); sw1, 10.1% ± 6.9% (1.2, 26.6)), one site in Ukraine (ukr7, 21.7% ± 14.1% (3.7, 57.9)), one site in Germany (ger1, 11.7% ± 9.3%

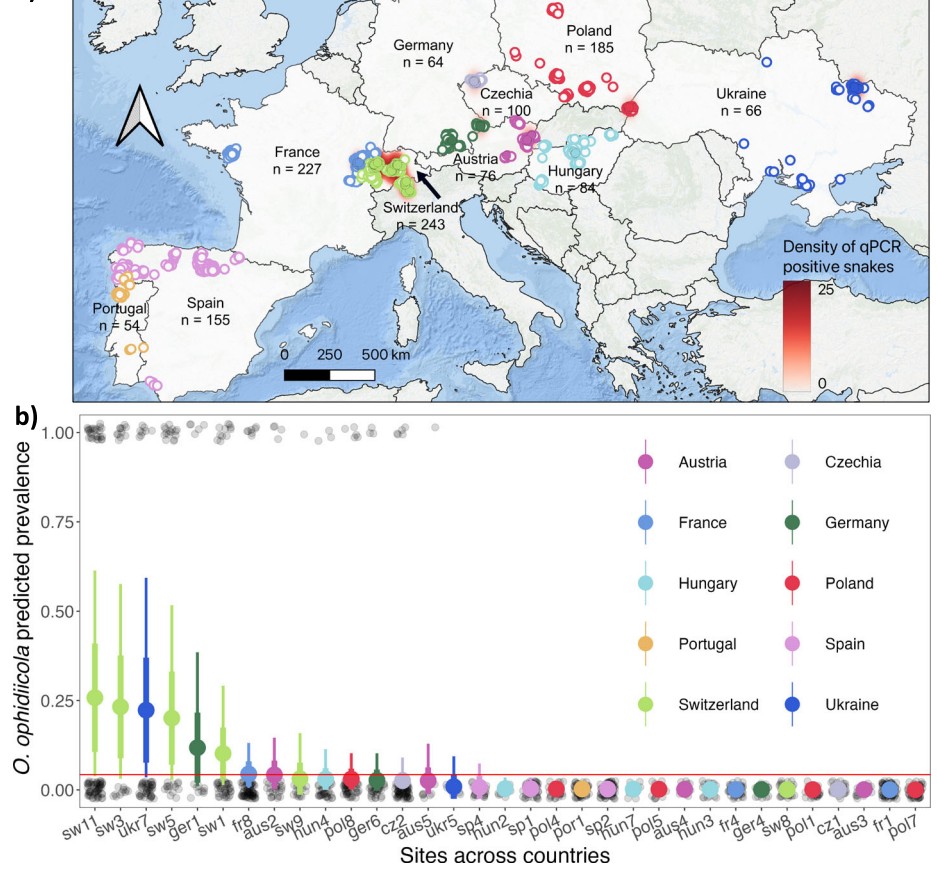

**Fig. 1 | Spatial distribution of snake captures and detections of *O. ophidiicola* across Europe. a** Each circle represents an individual snake capture and overlapping points were jittered for visualization. Different colors are used to distinguish countries, filled points indicate snakes that were qPCR positive (*n* = 109), and outlined points are qPCR negative snakes (*n* = 1145). Underlying density heatmap shows spatial distribution of *O. ophidiicola* infection risk based on qPCR positive detections using a kernel density estimation algorithm. We used a 100-km radius around each positive point and the scale bar indicates point density (i.e. relative disease risk) across each region. **b** Estimated *O. ophidiicola* prevalence across sites (*n* = 33) sampled in 10 countries. Each transparent black circle represents a single snake as being either negative (0) or positive (1). Larger circles and whiskers show the model predicted posterior mean ± standard deviation (thick lines), and 95% credible intervals (thin lines). Color of the circles represent the different countries; red line shows threshold for elevated prevalence defined as the estimated mean prevalence across all sites.

(1.0, 35.9)), and one in France (fr8, 4.4% ± 3.3% (0.6, 12.8)) (Fig. 1, Supplementary Table 3, for exact location of sites see Supplementary Table 2). The size of snakes sampled broadly overlapped among sites and we did not detect an effect of size on the probability of detecting the pathogen at a site (size coeff: 0.02 ± 0.01 (−0.00, 0.04)).

In addition to geographic variation, differences in *O. ophidiicola* prevalence were considerable among species (for observed prevalence values, see Supplementary Fig. 3). We found statistical support that species in the *Natrix*, *Hierophis* and *Zamenis* genera had higher *O. ophidiicola* prevalence (4.3%, *Natrix* intercept: −3.18 ± 0.60 (−4.58, −2.16); 5.0% and 2.6%, coeffs: 0.07 ± 0.59 (−1.11, 1.17) and −0.54 ± 0.45 (−1.45, 0.35) for *Hierophis* and *Zamenis*, respectively) than other genera sampled (*Coronella* coeff: −3.16 ± 1.26 (−6.06, −1.19); *Dolichophis* coeff: −8.43 ± 5.92 (−22.11, −0.31); *Vipera* coeff: −4.06 ± 0.96 (−6.06, −2.42)). There was also large variation in pathogen prevalence among congeneric species. *Natrix tessellata* had higher *O. ophidiicola* prevalence (model prediction: 9.0%; Fig. 2 and Supplementary Table 4) compared to several other members of the *Natrix* genus including *Natrix helvetica* (2.6%, coeff: −1.33 ± 0.66 (−2.69, −0.10), *Natrix astreptophora* (0.6%, coeff: −7.93 ± 6.02 (−22.23, 0.80)), and *Natrix maura* (0.1%, coeff: −8.79 ± 5.42 (−22.10, −1.40)), but there was no statistical support for differences compared to *Natrix natrix* (5.5%, coeff: −0.17 ± 0.78 (−1.72, 1.39)); Fig. 2 and Supplementary Table 4).

Overall, skin lesions were observed in 187 snakes from 15 species across all countries, but only 46.5% of those tested positive by qPCR ($n = 87$). Of all the snakes that tested positive by qPCR ($n = 109$), 80% of those had skin lesions that were consistent with SFD, while the other 20% had no visible skin lesion ($n = 22$) (Fig. 3a, Supplementary Fig. 4). We found a high probability of detecting lesions on a snake if they tested positive for *O. ophidiicola* (range 41.1%–91.7%, except for two viper species which had no visual sign of disease, Fig. 3a). We observed variation in disease severity among species but there was no statistical support for this relationship (Fig. 3b, Supplementary Table 5).

Genotyping analyses were successful for 85.3% of positive swabs (93 total swab samples) that were qPCR positive for *O. ophidiicola*. A total of four unique genotypes were observed, belonging to two of the major *O. ophidiicola* clades (clade I & II[41],) (Supplementary Table 6). Two of these genotypes (designated here as I-A and I-B) resided within clade I (i.e., the "European clade"). What we refer to as genotype I-A had an ITS2 sequence identical to strains previously isolated from Great Britain,

while genotype I-B had an ITS2 region sequence identical to a strain from Czechia[30]. The remaining two genotypes that we observed in our study resided within clade II (i.e., the "North American clade") and were identical to ITS2 region sequences of clonal lineages II-D/E (lineages D and E have identical sequences in the ITS2 region) and II-F[41]. Here we refer to these genotypes as II-D/E and II-F, respectively, although strains detected in this study may not be true representatives of the clonal lineages reported from North America since recombinant strains can have identical ITS2 sequences as clonal lineages[41]. Genotype I-A was detected primarily in western Europe (Switzerland, Germany, and Austria), whereas genotype I-B was detected in eastern Europe (Czechia, Austria, Hungary, Poland, and Ukraine) (Fig. 4a). Genotype II-D/E was more widely distributed across Europe, whereas genotype II-F was only found along a single lake in Switzerland (Fig. 4a, inset). On two occasions, snakes were found to be infected with multiple genotypes of *O. ophidiicola*: a snake from Switzerland from which genotypes I-A and II-D/E were detected and a snake from Hungary from which genotypes I-A and I-B were detected.

The top two models, as determined through LOO, that best predicted *O. ophidiicola* prevalence across the landscape included both host species and pathogen clade as predictor variables, with the best fit model including an interaction between these two variables (Supplementary Table 7). There was statistical support that *Z. longissimus* had a higher probability of being infected when clade II was detected at a site compared to just clade I (coeff: 2.58 ± 1.12 (0.56, 4.93)). Conversely, we found that *N. helvetica* was associated with higher probability of infection when clade I was detected at a site (coeff: −1.25 ± 1.21 (−3.62, 1.22) (Fig. 4b, Supplementary Table 8). Other species, *N. natrix and N. tessellata*, also had a higher probability of pathogen detection when clade II was detected at a site, but the credible intervals included zero (*N. natrix* coeff: 0.78 ± 1.04 (−1.34, 2.78), *N. tessellata* coeff: 1.15 ± 0.85 (−0.58, 2.82)). Snakes that were infected by a strain of *O. ophidiicola* belonging to clade I generally had less severe disease when compared to snakes infected with a strain from clade II, although the credible intervals overlapped zero (clade I coeff: −0.02 ± 0.34 (−0.57, 0.52), Fig. 4c, d and Supplementary Fig. 5).

## Discussion
Our results support the presence of hotspots for *O. ophidiicola* across Europe and identify factors that may contribute to higher infection prevalence at a site. *O. ophidiicola* was detected in all countries studied except

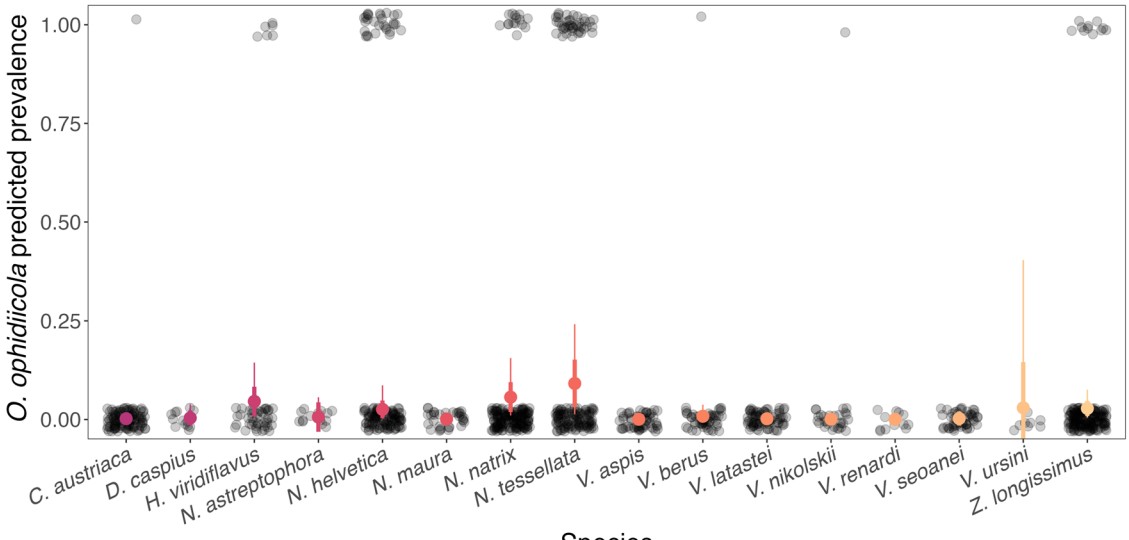

**Fig. 2 | Predicted prevalence of *O. ophidiicola* in different species of snakes across Europe.** Each transparent black circle represents a single snake as being either negative (0) or positive (1). Larger circles and whiskers show the model predicted posterior mean ± standard deviation (thick lines), and 95% credible intervals (thin lines) for each species across all countries. Colors indicate different species ($n = 16$).

**Fig. 3 | Lesion prevalence and disease severity in *O. ophidiicola*-positive snakes across different species.** Color circles and whiskers show the model predicted posterior mean, ±standard deviation (thick lines), and 95% credible intervals (thin lines) for different species across all countries. **a** Each transparent black circle represents a single snake as being either negative (0) or positive (1) for presence of lesions, which was used to calculate the proportion of the population that tested positive (prevalence). **b** Each transparent black circle represents the percentage of the body of a single snake covered in lesions and the size of the circle is proportional to the total surface area of the snake (scale ranges 250–1000 cm²).

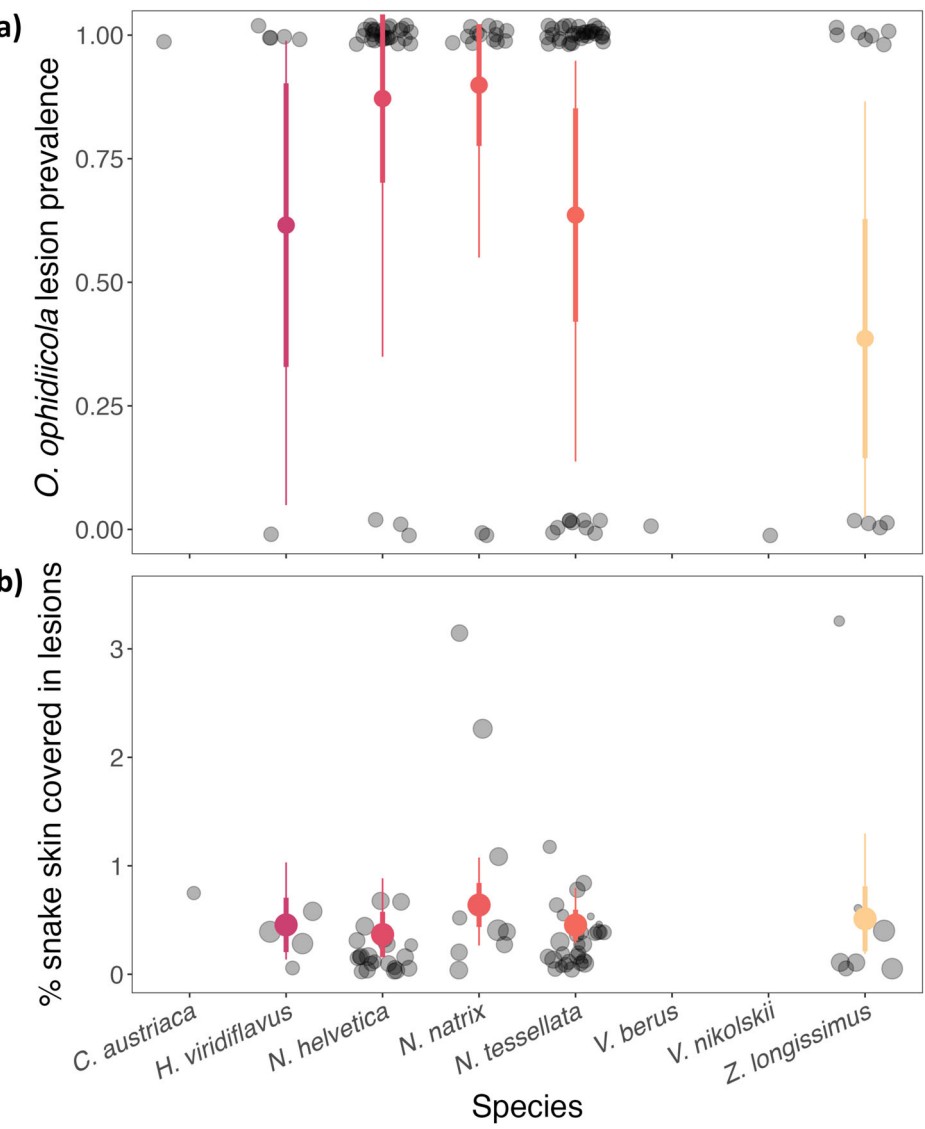

for those on the Iberian Peninsula (Spain and Portugal), which may be attributed to numerous factors (e.g. different species community, geographical barriers such as Pyrenees mountains, or environmental conditions) that may be unsuitable for persistence and growth of *O. ophidiicola*. It is also possible that the pathogen has not spread to this area. Our surveys establish seven sites across four countries with elevated prevalence of *O. ophidiicola* in Europe, with the highest being in Switzerland. We find that the host species and pathogen clade present at a site were important for predicting areas with elevated prevalence. Environmental factors (climate, proximity to water, pollution, etc.) could also contribute to variation in pathogen prevalence. However, environmental features associated with sites that had high prevalence broadly overlapped with sites that had low prevalence or even no pathogen detection, suggesting these are unlikely to be the most important factors in this system.

Snake fungal disease has garnered much attention over the last few decades, as this disease has been recognized as a potential threat to snake populations[28,36]. Despite this, few studies have systematically examined geographic differences in prevalence and disease severity and the contribution of host pathogen interactions. We found no seasonal variation in *O. ophidiicola* prevalence, which is contrary to several previous studies[38,45,46]. Instead, we found high variability of infection prevalence among snake species, which has also been described in other regions[45,47]. Underlying host characteristics such as dependence on aquatic habitats, have previously been found to be

associated with higher *O. ophidiicola* infection prevalence[38], which could partly explain increased infection in the *Natrix* genus. *Natrix tessellata* had the highest pathogen prevalence, followed by *N. natrix* and *N. helvetica*, with all three species being either semiaquatic or living near water (with *N. tessellata* being more piscivorous). Interestingly, only two vipers (out of a total of 341 samples) tested positive for *O. ophidiicola*, and both snakes had no visual signs of infection (i.e. no lesions present). This indicates that viperids may not be competent hosts for *O. ophidiicola*, possibly due to innate behavioral and physiological mechanisms or environmental association.

In our study, no mortality was reported, and snakes generally appeared healthy except in a few cases where infection was severe and had spread to the face with possible disruption to foraging behavior. The low prevalence and disease severity observed in Europe could be the result of lower pathogen virulence or decreased host susceptibility. We also found that only 46% of snakes with lesions tested positive for *O. ophidiicola*, which has also been reported from North America[47,48]. The lesions that could not be attributed to *O. ophidiicola* infection looked similar to SFD skin lesions and may be fungal or bacterial in origin[49], suggesting the need to investigate other sublethal effects of SFD and the interaction between *O. ophidiicola* and other pathogens. Alternatively, it is possible that snakes with lesions that tested negative were false negatives perhaps because snakes had recently shed, and the pathogen load was too low to detect by qPCR. Collection of tissue samples could help resolve this possibility.

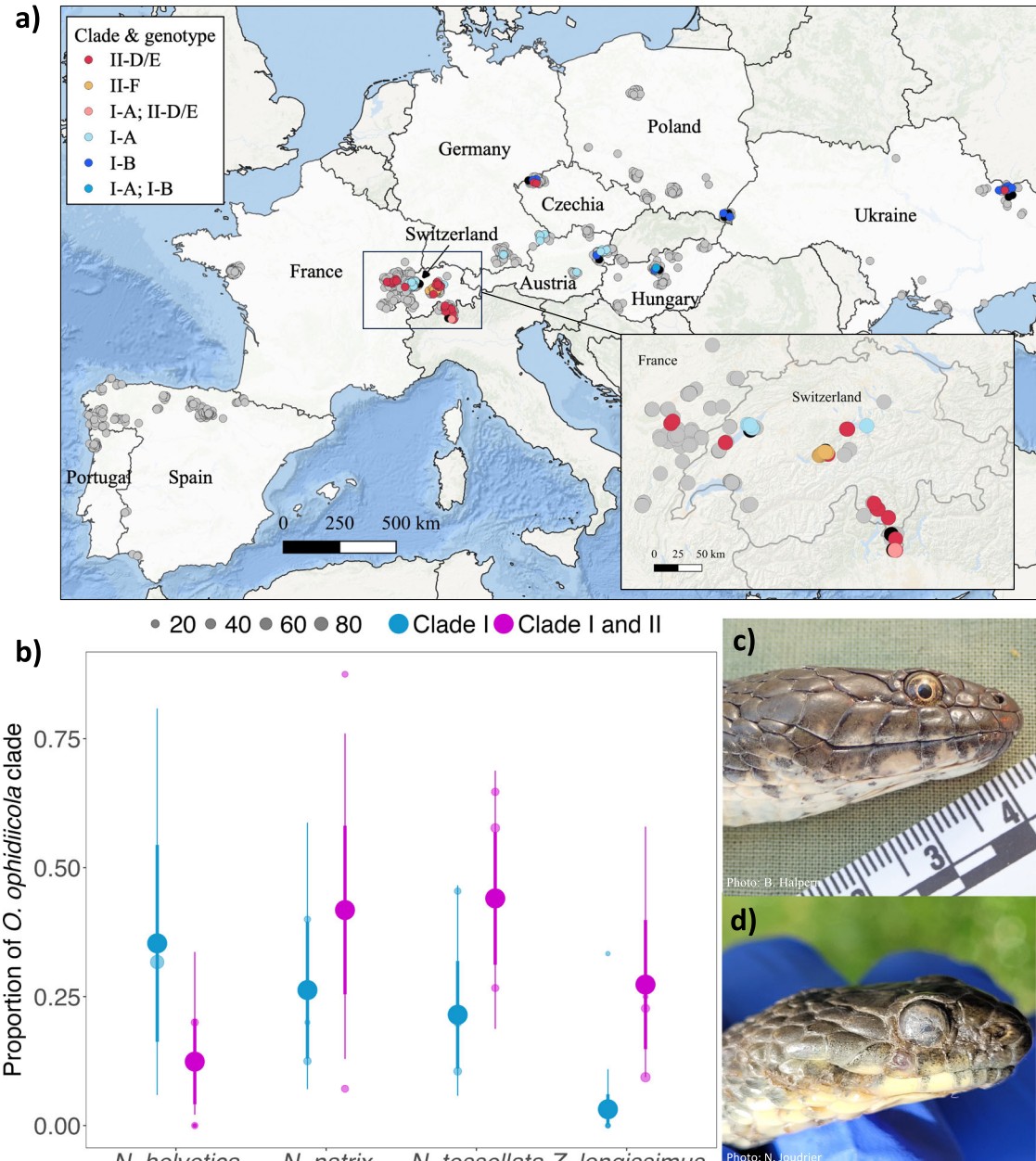

**Fig. 4 | Distribution of *O. ophidiicola* clades and genotypes across Europe. a** *O. ophidiicola* clades and genotypes from positive detections (n = 93) across the landscape in Europe. Points are slightly jittered for visual representation of the sampling range. Color of the point indicates the clades and genotypes, samples that were qPCR negative are represented as gray points (n = 1145), and samples that failed to amplify with the genotyping PCR are represented as black points (n = 16). Pink and medium blue points represent simultaneous detections of genotypes I-A and II-D/E and genotypes I-A and I-B, respectively, from the same swab sample (i.e., snakes infected with multiple genotypes). The enlarged map (inset) shows better resolution of detections in Switzerland. **b** The proportion of each *O. ophidiicola* clade (clade I or clade I & II) detected by species. Small color points are predicted mean prevalence at a site for a given species and clade, point size indicates sample size at each site. Large color points and whiskers show the model predicted posterior mean, ±standard deviation (thick lines), and 95% credible intervals (thin lines) across different species. **c** Photo of a *N. tessellata* from Hungary infected with *O. ophidiicola* from clade I-B. **d** Photo of a *N. tessellata* from Switzerland infected with *O. ophidiicola* from clade II-F showing facial infection.

We found that models accounting for both host species and pathogen clade best explained the variation in pathogen prevalence across the landscape. Importantly, the top model included an interaction between host species and pathogen clade, indicating that the effect of clade is not the same across species. We found that when clade II was present in an area, there was support that the probability of detecting *O. ophidiicola* was higher for three (*N. natrix*, *Z. longissimus*, and *N. tessellata*) of the four species with the highest prevalence. Hotspots in Switzerland were primarily driven by positive detections in *N. tessellata*, and we detected *O. ophidiicola* belonging to clade II in three of the four hotspots in

Switzerland, indicating the importance of both the presence of specific host species and pathogen clade when trying to understand drivers of disease patterns.

The history and origin of *O. ophidiicola* in Europe is unclear. Sampling of museum specimens has revealed that strains of *O. ophidiicola* belonging to clades I and II were present in Switzerland as early as 1959[44]. It is estimated that clade I shared a common ancestor within the last 100–500 years, but that analysis only included four clade I strains and may greatly underestimate the time that *O. ophidiicola* has been present in Europe[41]. Despite the designation of clade II as the "North American" clade, it is believed that *O. ophidiicola* is

not native to North America and multiple introductions (most likely from Eurasia) have occurred over the last century[41]. Thus, it is plausible that either or both clades I and II are native to Europe. Strains of *O. ophidiicola* isolated from wild snakes in Taiwan reside within clade II[32,41], which could also indicate a southeast Asian origin for that clade, raising the possibility that clade II is not native to Europe. We detected two genotypes within clade II. One of these (II-D/E) was widely distributed, whereas the other (II-F) was detected in a single snake community around a lake in Switzerland. That snake community includes an introduced population of *N. tessellata*. Taken together, this could indicate that genotype II-F was more recently introduced to Europe, perhaps through the release of snakes originating in captivity. However, determining the genetic diversity and origin of the various lineages of *O. ophidiicola* in Europe would require more in-depth studies.

In conclusion, we find several pathogen hotspots in Europe, which could be attributed, at least partially, to specific host species and the presence of particular pathogen strains. This relationship varied, and in some cases, the effect of pathogen genotypes was reversed for some species, indicating that just pathogen identity alone cannot explain the observed landscape patterns. Although virulence is recognized as an important factor in the effects of disease on host populations, the general lack of landscape level data on pathogen strain distribution and association with disease has likely limited our ability to determine its importance for other disease hotspots.

## Methods

### Location and host species considered
Free-ranging snakes were captured from March 2020 to June 2022 across 10 countries: Portugal, Spain, France, Switzerland, Germany, Austria, Czech Republic, Hungary, Poland, and Ukraine. The number of sites where snakes were collected ranged from two to eleven per country, for a total of 61 sites. Sites were selected based on pre-existing knowledge of snake presence or prediction of suitable habitats to obtain sufficient sample sizes across a range of species.

### Capture and sampling
Handling of snakes was reviewed and approved by Virginia Tech Institute for Animal Care and Use Committee protocol 20-055 and permits to conduct our field study were obtained when necessary. We have complied with all relevant ethical regulations for animal use.

Snakes were located by visual encounter surveys, an approach frequently used for sampling snakes[50]. Snakes were captured by hand, placed in individual cloth or paper bags for temporary holding during processing and sampling, and released at their capture location. Sterile handling procedures were followed (disposable gloves between each snake, individual sterilized bags or disposable paper bags, and equipment was cleaned between snakes using 70% ethanol) during sample collection to avoid cross-contamination. Snakes were individually identified using photo-identification or marking (using PIT tags or scale-clipping). For each snake captured, location and morphometric data were collected including latitude/longitude, species, sex, snout-vent-length, tail length, and weight. In addition, we recorded if snakes had skin lesions present on their body, and for the snakes that had visible lesions, we collected photos to quantify disease severity (see below).

Snakes were swabbed in duplicate (except for a few individuals that were swabbed only once due to limitations in the field) using a pre-moistened (using sterile water), sterile polyester-tipped applicator (Puritan®, Guilford, Maine, USA) by running the swab five times (back and forth counting as a single pass) on the ventral and dorsal areas from the neck down to the vent, and two times on the face of the snake. If a skin lesion was observed, a separate swab was used to specifically swab the lesion and skin immediately adjacent to it by rubbing the swab over the affected skin. Swabs were individually stored in a 2 mL sterile tube in a cooler with ice while in the field and stored frozen at −20 °C in the lab until analysis.

### Sample extraction and qPCR
Swab samples were collected and processed by one of two laboratories using the same methods. DNA was extracted from swabs using 250 μL of PrepMan® Ultra Sample Preparation Reagent (Life Technologies, Carlsbad, California, USA) with 100 mg of zirconium/silica beads, following a previously published protocol[51]. Briefly, samples were homogenized for 45 s in a bead beating grinder and lysis system (MP Biomedicals, Irvine, California, USA) and centrifuged for 30 s at 13,000 × *g* to settle all material to the bottom of the tube. Homogenization and centrifugation steps were repeated, and tubes were incubated at 100 °C in a heat block for 10 min. Tubes were then cooled at room temperature for 2 min, then centrifuged for 3 min at 13,000 × *g*. Fifty to 100 μL of supernatant was recovered and stored at −80 °C. Extraction blanks (negative controls to ensure no contamination occurred during extraction) were prepared using 250 μL of PrepMan Ultra Sample Preparation Reagent and 100 mg of zirconium/silica beads only.

Quantitative PCR targeting the internal transcribed spacer region (ITS) specific to *O. ophidiicola* was performed on a real-time PCR QuantStudio 5 (Thermofisher Scientific, Waltham, Massachusetts, USA)[52]. QuantiFast Master Mix (QuantiFast Probe PCR + ROX vial kit, Qiagen, Germantown, USA) was prepared according to manufacturer's recommendations for a final reaction volume of 25 μL, which included 5 μL of extracted DNA. Cycling conditions were as follows: 95 °C for 3 min, then 95 °C for 3 s and 60 °C for 30 s for a total of 40 cycles. For each plate run, a negative control (water added instead of extracted DNA to ensure no contamination occurred during PCR) and a 6-point (each point run in triplicate) standard curve using synthetic double-stranded DNA (gBlock, Integrated DNA Technologies, Coralville, Iowa) of the target region ($1.0 \times 10^2$, $1.0 \times 10^1$, $1.0 \times 10^0$, $1.0 \times 10^{-1}$, $1.0 \times 10^{-2}$, $1.0 \times 10^{-3}$ fg/μL) were included. Samples that were positive were analyzed in duplicate, and a snake was determined to be *O. ophidiicola* positive if any swab associated with that snake was positive by qPCR (regardless of clinical signs being present or not). Based on the limit of detection from the PCR assay, the Ct threshold was set at 39 for both labs.

### Sequencing and genotyping
Samples in which *O. ophidiicola* was detected by qPCR in at least one swab were subjected to follow up genotyping analysis. We targeted a portion of the internal transcribed spacer 2 (ITS2) for this analysis because the ITS2 exhibits variability between previously described clades of *O. ophidiicola* and because ITS2 is a multicopy gene that can be amplified from samples containing very small amounts of *O. ophidiicola* DNA (it is also the target of the qPCR assay). We used a nested PCR protocol that consisted of first amplifying the entire ITS2 region with the panfungal primer ITS3 and ITS4[53]. The first reaction consisted of 10 μL of 2× QuantiNova probe PCR master mix (Qiagen, Venlo, Netherlands), 3.9 μL of molecular grade water, 0.5 μL of each primer (20 μM each), 0.1 μL of 20 μg/μL bovine serum albumin, and 5 μL of DNA extracted with the PrepMan procedure described above. Cycling conditions were as follows: 95 °C for 3 min; 40 cycles of 95 °C for 10 s, 56 °C for 30 s, and 72 °C for 30 s; final extension at 72 °C for 5 min. For the second reaction, primers ITS3 and Oo-rt-ITS-R[52] were used. Each reaction consisted of 0.5 μL of the PCR product from the first reaction added to 13.375 μL molecular grade water, 5 μL of GoTaq Flexi buffer (Promega Corporation, Madison, Wisconsin, USA), 2 μL of dNTPs (2.5 mM each), 1.5 μL of 25 mM MgCl$_2$, 1.25 μL of each primer (20 μM each), and 0.25 μL of GoTaq polymerase. Cycling conditions for the second PCR were: 95 °C for 10 min; 45 cycles of 95 °C for 30 s, 56 °C for 30 s, and 72 °C for 1 min; final extension at 72 °C for 5 min. Products from the second PCR were visualized on an agarose gel, and those containing bands were sequenced in both directions using the Sanger method with primers ITS3 and Oo-rt-ITS-R.

Samples that generated messy chromatograms or appeared to contain single nucleotide polymorphisms (SNPs) indicative of multiple *O. ophidiicola* genotypes were re-amplified with the second PCR using a proofreading polymerase (15.75 μL of molecular grade water, 5 μL of 5× SuperFi buffer [Thermo Fisher Scientific Corporation, Waltham, Massachusetts, USA], 2 μL of dNTPs (2.5 mM each), 0.625 μL of 20 μM each primer, 0.5 μL of Platinum SuperFi DNA polymerase [2U/μL], and 0.5 μL of product from the first PCR; cycling conditions were the same as described for the second reaction above). The resulting amplicons were cloned using the Invitrogen Zero Blunt TOPO PCR cloning kit for sequencing (Thermo

Fisher Scientific Corporation, Waltham, Massachusetts, USA), and individual transformants were sequenced.

Individual ITS2 sequences generated in our study were assigned to genotypes. Sequences with 100% identity across the ITS2 region of *O. ophidiicola* were considered to be the same genotype; any sequence differing from another by at least one SNP was classified as a unique genotype.

## Quantification of disease severity

Disease severity was measured by calculating the fraction of surface area of each snake covered by lesions. Using the image processing program ImageJ[54] and the photos of the snakes taken in the field, we measured each lesion five times and recorded the mean length and width. We calculated the surface area of each lesion on a particular snake (i.e., length × width) and added up the surface area of each lesion present on a snake to determine the total lesion surface area for that snake. Using the morphometric measurements collected from each individual, we also calculated the total surface area for each snake (snout to cloaca was treated as a cylinder, cloaca to tip of the tail as a cone), to estimate the percentage of total surface area covered by lesions.

## Statistics and reproducibility

We present analytical methods in the order that they appear in the Results. In addition, we included a supplementary table (Supplementary Table 1) describing all the statistical analyses, whether the results are displayed in a figure or a table and the parameters of the model.

We first examined whether there was seasonality associated with *O. ophidiicola* detection by running a Bayesian multilevel model with a Bernoulli distribution and a logit link where the detection of *O. ophidiicola* for each individual snake (0|1), was our response variable, Julian date of each capture date of a snake was our population-level effect (i.e. predictor variable) and species and sites were our group-level effects (i.e. random effect). We then compared prevalence of *O. ophidiicola* among sites to identify areas with elevated pathogen prevalence across Europe based on pathogen detection using qPCR. We used a Bayesian multilevel model with a Bernoulli distribution and a logit link where the detection of *O. ophidiicola* for each individual snake (0|1), was our response variable, site was our population-level effect and species was our group-level effect. To detect sites with elevated pathogen prevalence (defined as 'hotspots') we identified *O. ophidiicola* estimated prevalence of sites that were greater than a chosen threshold[55]. The chosen threshold was the estimated mean *O. ophidiicola* prevalence across all sites. We also examine the upper 25% quantiles of the data which yielded qualitatively similar results. We investigated if snake size influenced *O. ophidiicola* prevalence using a Bayesian multilevel model with a beta distribution and logit link with the estimated prevalence at each site from the previous model as our response variable, the average SVL for each species at a site as our predictor variable, and a group-level effect of species.

To investigate the effect of host genus on pathogen prevalence, we used a Bayesian multilevel model with a Bernoulli distribution and logit link with pathogen detection for each snake as our response variable, genus of the snake being sampled as our predictor variable, and a group-level effect of site. We also examined differences in pathogen prevalence among host species, using a similar model to what is described above, but we included snake species as our predictor variable, and a group-level effect of site.

To investigate differences in lesion prevalence in snakes that were qPCR positive, we first ran a Bayesian multilevel model with a Bernoulli distribution and logit link, with the detection of lesion (0|1) as our response variable, species as our predictor variable, and a group-level effect of site. We also examined differences in disease severity for the snakes that were positive for *O. ophidiicola*, with a beta distribution. Our response variable was the percentage of total surface area of the snake covered in lesion, with species as our predictor, and site our group-level effect.

We examined the best model that explained *O. ophidiicola* prevalence using leave-one-out cross-validation (LOO). The Bayesian multilevel models that were tested included the population level effects of just species,

just pathogen clade, an additive model of species and clade, and an interactive model of species and clade, with a Bernoulli distribution, pathogen detection as our response variable (0|1), and site as a group-level effect for all four models. Models were run with a total of 4 chains for 6000 iterations each, with a burn-in period of 1500 iterations per chain resulting in 18,000 posterior samples. For the variable "pathogen clade", we pooled the four genotypes (i.e., I-A, I-B, II-D/E and II-F) into their respective clade (clade I and II) as sample sizes were generally too small across species to look at genotypes separately. The clade variable consisted of either clade I (sites where only snakes infected with clade I were detected) or both clades I and II (sites where snakes were infected with *O. ophidiicola* belonging to clade I or clade II), as clade I is widely distributed across sites and there were only a few locations with detections of only clade II. Prevalence by pathogen clade comparisons is only reported for the four species with the highest prevalence (*N. natrix*, *N. tessellata*, *N. helvetica*, and *Z. longissimus*) for which there was sufficient data. The final clade dataset used for this analysis included 14 sites and 558 snakes. Finally, to examine how pathogen clade influenced infection severity, we performed an analysis with a beta distribution where our response variable was the percentage of total surface area of the snake covered in lesion, and our predictor variable was pathogen clade, with a group-level effect of species.

We fit all models (unless otherwise noted) using the No-U-Turn Sampler (NUTS), an extension of Hamiltonian Markov chain Monte Carlo (HMCMC). All Bayesian models were created in the Stan computational framework (http://mc-stan.org/) accessed with the "brms" package[56]. To improve convergence and avoid over-fitting, we specified weakly informative priors (a normal distribution with mean of zero and standard deviation of 10). Models were run with a total of 4 chains for 2000 iterations each, with a burn-in period of 1000 iterations per chain resulting in 4000 posterior samples, which, given the more efficient NUTS sampler, was sufficient to achieve adequate mixing and convergence. All $\hat{R}$ values were less than or equal to 1.01, indicating model convergence. For all analyses, we excluded any species and sites that had been sampled fewer than eight times (site sample size = 33, species sample size = 16), and we assessed statistical support using credible intervals that do not overlap zero. Statistical analyses were performed in R software version 4.2.0[57].

## Reporting summary

Further information on research design is available in the Nature Portfolio Reporting Summary linked to this article.

## Data availability

The source data of all figures is available in Supplementary Data 1.

## Code availability

The code generated in this study is available upon reasonable request to the corresponding author.

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

## Acknowledgements
We thank Otto Aßmann, Florian Bacher, Jacek Błachuta, Philomin Briot, Jon Buldain, Maxime Chèvre, Pierre Cheveau, Sylvain Dubey, Anna Egerer, Lea Endrejat, Karin Ernst, Niklas Franzen, Inês Freitas, Georg Gassner, Davy Guinchard, Krisztián Harmos, Marietta Hengl, Piotr Hubal, Kacper Jurczyk, Yurii Kornilev, Roman Kurek, Nahla Lucchini, Adrian Neumann, Daniel Renner, Sandra Wallner, Bartłomiej Zając, and the team at the Museum of Besançon (France) for assistance in the field. Many thanks also to Kate Langwig for statistical insights, and to Oleksandra Klynova and Megan Winzeler for laboratory assistance. This material is based upon work supported by the National Science Foundation Graduate Research Fellowship under Grant No. 480040. Additional funding to G.B. was provided by a Virginia Tech Cunningham fellowship, and a CeZAP (Center for Emerging, Zoonotic, and Arthropod-borne Pathogens) grant as part of the Infectious Diseases (ID) Interdisciplinary Graduate Education Program (IGEP). F.M.-F. is supported by FCT - Fundação para a Ciência e a Tecnologia, Portugal (contract ref. DL57/2016/CP1440/CT0010). Sampling in Poland was funded by the statutory funds of the Institute of Nature Conservation, Polish Academy of Science. Sampling in Hungary was supported by Duna-Ipoly National Park, Duna-Dráva National Park, Aggtelek National Park, and LIFE HUNVIPHAB project (LIFE18NAT/HU/000799). Sampling in France was funded by Voies Navigables de France, the Agence de l'eau Rhône-Méditerranée-Corse, Région Bourgogne-Franche-Comté, DREAL Bourgogne-Franche-Comté, Département du Jura, Doubs and Territoire de Belfort, and UNICEM (to A.M. and T.C.), and by the Regional Council of Nouvelle-Aquitaine and Aquastress project (2018-1R20214 to O.L.). Any use of trade, firm, or product names is for descriptive purposes only and does not imply endorsement by the U.S. Government.

## Author contributions
G.B.: conceptualization, data curation, investigation, formal analysis, visualization, methodology, writing—original draft, writing—review and editing; J.M.L.: methodology, investigation, writing—review and editing; N.J.: methodology, resources, investigation, writing—review and editing; S.B.: investigation, resources, writing—review and editing; T.C.: methodology, investigation, writing—review and editing; M.F.: investigation, writing—review and editing; F.M.-F.: investigation, writing—review and editing; G.G.: investigation, writing—review and editing; B.H.: investigation, writing—review and editing; A.K.: investigation, writing—review and editing; K.K.: investigation, writing—review and editing; O.L.: investigation, writing—review and editing; A.M.: methodology, investigation, writing—review and editing; R.M.: investigation, writing—review and editing; S.S.: investigation, review and editing; B.S.: investigation, writing—review and editing; S.U.: methodology, resources, investigation, writing—review and editing; O.Z.: investigation, writing—review and editing; J.R.H.: conceptualization, resources, supervision, funding acquisition, writing—review and editing.

## Competing interests
The authors declare no competing interests.

## Additional information

¹Biological Sciences Department, Virginia Polytechnic Institute and State University, Blacksburg, VA, USA. ²U.S. Geological Survey, National Wildlife Health Center, Madison, WI, USA. ³Institute of Biology, University of Neuchâtel, Neuchâtel, Switzerland. ⁴Institute of Animal Pathology, University of Bern, Bern, Switzerland. ⁵Info fauna-Karch, Centre Suisse de Cartographie de la Faune (CSCF) and Centre de coordination pour la protection des reptiles et des amphibiens de Suisse (karch), Neuchâtel, Switzerland. ⁶Department of Comparative Anatomy, Institute of Zoology and Biomedical Research, Jagiellonian University, Cracow, Poland. ⁷NATRIX Herpetological Association, Wroclaw, Poland. ⁸LPO Bourgogne-Franche-Comté, Site de Franche-Comté, Maison de l'environnement de BFC, Besançon, France. ⁹Bavarian State Collection of Zoology (ZSM-SNSB), Munich, Germany. ¹⁰CIBIO, Centro de Investigação em Biodiversidade e Recursos Genéticos, InBIO Laboratório

Associado, Campus de Vairão, University of Porto, Vairão, Portugal. [11]BIOPOLIS Program in Genomics, Biodiversity and Land Planning, CIBIO, Campus de Vairão, Vairão, Portugal. [12]Le Grand Momesson, Bouvron, France. [13]MME BirdLife Hungary, Budapest, Hungary. [14]Department of Systematic Zoology and Ecology, Institute of Biology, Eötvös Loránd University, Budapest, Hungary. [15]HUN-REN-ELTE-MTM, Integrative Ecology Research Group, Budapest, Hungary. [16]Department of Geoinformatics and Cartography, Institute of Geography and Regional Development, Faculty of Earth Sciences and Environmental Management, University of Wroclaw, Wroclaw, Poland. [17]Department of Wildlife Conservation, Institute of Nature Conservation Polish Academy of Science, Cracow, Poland. [18]Centre d'Etudes Biologiques de Chizé, ULR CNRS UMR 7372, Villiers en Bois, France. [19]School of Life Sciences, Arizona State University, Tempe, AZ, USA. [20]Zamenis Civic Association, Karlovy Vary, Czech Republic. [21]First Zoological Department, Herpetological Collection, Natural History Museum, Vienna, Austria. [22]Department of Genetics, Kazimierz Wielki University, Bydgoszcz, Poland. [23]Department of Environmental Sciences, Section of Conservation Biology, University of Basel, Basel, Switzerland. [24]Balaton Limnological Research Institute, Tihany, Hungary. [25]V.N. Karazin Kharkiv National University, Kharkiv, Ukraine. ✉e-mail: gaelle.blanvillain@gmail.com

