## [Peer review file · Communications Biology]

Reviewers' comments:

Reviewer #1 (Remarks to the Author):

This manuscript describes a broad survey for ophidiomycosis in free-ranging snakes in Europe. The authors tested snakes for *Ophidiomyces ophidiicola* using qPCR of swabs, noted the presence and size of lesions, and conducted genotyping on the *O. ophidiicola* isolates. They found that geographic hotspots for *O. ophidiicola* existed in Switzerland, Ukraine, Germany and France, certain species of snakes were more likely to be infected with *O. ophidiicola*, and multiple clades of *O. ophidiicola* were present among sampled snakes, with *O. ophidiicola* prevalence at particular sites best predicted by snake species and pathogen clade.

Overall, this work represents a significant contribution to the understanding of ophidiomycosis in Europe. I commend the authors on addressing not only skin lesion and pathogen detection, but also measurement of clinical disease severity and genomic analysis. The manuscript is well-written with clear goals, methods, and results. The work could be reproduced based on the level of detail provided and the conclusions adequately supported by the findings. The figures are high-quality and well-explained.

Specific comments:

1. Be careful not to use the terms "disease," "infection," and "pathogen" interchangeably. Disease indicates the presence of clinical signs when the host responds to the presence of the pathogen, infection indicates the presence/growth of the pathogen in/on the host, and pathogen is the infectious agent. For example, on line 54 you refer to "disease hotspots of *Ophidiomyces ophidiicola*", but *Oo* is the pathogen, not the disease. Is this paper addressing disease or prevalence hotspots?
2. Line 86: Suggest adding "the" between "theory suggests that" and "presence"
3. Lines 104-105: Suggesting adding a citation for Pribyl et al 2023 (*Oo* in snakes in the Czech and Slovak Republics, *Journal of Vertebrate Biology*, DOI: 10.25225/jvb.23050)
4. Lines 210, 258: Are these the same genotypes and clades defined in the Ladner et al 2023 paper?
5. Figure 1a: "n" is missing in the sample size label for Spain (currently reads "Spain = 155").
6. Several Figures (1b, 2, 3): Very nice figures. The figure legends mention "black circles" for individual snakes testing positive or negative, but the circles appear gray in the figures.
7. Figure S4: Photos are described as "mild", "moderate" and "severe" in the figure legend, but these subjective terms have not been previously defined in the manuscript. Suggest leaving out the subjective descriptions.
8. Line 456: Consider rephrasing to "hotspots for *Ophidiomyces ophidiicola*" to more accurately reflect your results.

Reviewer #2 (Remarks to the Author):

This is an extremely interesting and timely study and I enjoyed reading it. I have concerns about the accuracy of the study framework and about some of the methods, but these would be simple to address in revisions. The study is worth polishing up – it is already impressive and will have greater impact with a few clarifications. I have made suggestions below, which I hope will be helpful.

General comments:

The likely impacts of ophidiomycosis are overstated, cherry-picking the literature to support an inaccurate "crisis" narrative. This is not helpful to our understanding of wildlife health, or to snake conservation. There is no need to overstate things: this disease is important to study even if it isn't "the next chytrid or WNS". Can you revise to present a more evidence-based perspective on the disease?

Throughout the paper: consider using "ophidiomycosis" instead of "snake fungal disease". SFD is an extremely vague name and there are a number of fungal pathogens that can cause disease in snakes. It is more correct (and recently, more common in the literature) to use the more specific term.

Consider not only susceptibility and resistance, but also tolerance to pathogens in your framework. Day of year for sample collection should be explicitly included as a factor in the analyses, in case the observed variation in prevalence is (partly or wholly) temporal, and not only spatial. Snake size (for example, using SVL) should be accounted for in analysis. Body size can affect detection probability for pathogens, because swabbing a larger surface area can pick up more fungal DNA.

Somewhere, you should note that this study did not employ full diagnostic criteria for the disease. So, it speaks to detected pathogen presence and to observed lesions consistent with ophidiomycosis, but did not confirm clinical ophidiomycosis.

Specific comments:

Lines 86-91: there is a third option here, supported by both theory and empirical evidence. If a species is tolerant to a pathogen (but not resistant), it can act as a reservoir host without experiencing population declines. This is how the fungus that causes WNS in bats persists in European bat populations. Some snakes in North America may also be tolerant to *Ophidiomyces*. For example, wild queensnakes do not exhibit reduced short-term survival (<https://esajournals.onlinelibrary.wiley.com/doi/abs/10.1002/eap.2251>) and wild eastern foxsnakes often recover from the disease during the active season (<https://doi.org/10.7589/JWD-D-21-00134>). Consider tolerance as an option, and revise other relevant pieces of the manuscript accordingly.

Lines 95-97: Ophidiomycosis was proposed as a serious threat to wild snakes in the papers cited, and others, but the aggregated data since then do not support this. Lorch et al (2016) is cited in Lines 99-100 without enough context. This study described a handful of population declines that were complex, with which the disease was associated but may not have been the main cause (as elegantly stated in that paper). You are not providing a balanced view of the data so far. In the previous line you cite Davy et al. (2021) to add to the list of affected snake species, but that study also summarized the evidence for impacts of ophidiomycosis on wild populations, and did not find support for the hypothesis that it is a serious threat. This could change in future as more populations are studied - but it seems appropriate to follow the available evidence for now.

Line 102 – delete “only”

Line 123 – the meaning of “we combined host species and pathogen clade” is unclear. It becomes more clear in the methods, but consider revising here to help the reader understand things earlier.

Methods:

This is impressive sampling and I applaud the team for coordinating this effort.

Please clarify the following details, to ensure the methods could be replicated (or that variation among studies can be taken into account):

- How were bags and equipment sterilized between snakes?
- Were gloves worn to handle snakes? If so, were they changed between snakes? If not, how did you avoid cross-contamination of samples when handling snakes?
- What was used to moisten the tip of the applicators?
- Prevalence of *Ophidiomyces* varies seasonally in North America. Please provide sampling dates for the sites, and consider incorporating ordinal date as a factor in your analyses.
- Lines 152-153 – can you confirm the meaning of “later”? Does this mean the cooler with ice and samples went directly to a lab and then a freezer, or was there different storage in between?
- Line 178: “positive by qPCR” means the sample had a Ct value below some threshold, selected based on the standard curve. Can you specify the Ct threshold used? Was it identical between the two labs? (N.B. – this is not intended to imply that results from the two labs were not comparable. It is simply important information to include regardless of what the details turn out to be. This also applies to line 181- what criteria were used for “detection”? Was it different between the two labs?

- Lines 165 and 173 describe two different types of negative controls that test for different potential sources of contamination (extraction blanks, and water). Were both negative controls run on the plates?

- Lines 221-223: it is not clear how you calculated surface area for snakes. Did you treat them as a cylinder? This wouldn't be quite accurate, but it is justifiable if that's what you did- it would give you a comparable SA per snake. Or did you do something different?

Line 144- should be "lesions"

Line 157: could revise to "using the following methods".

Lines 215-223 – why did you not also consider the number of lesions as a measure of disease severity?

Lines 226-228: this text is odd. Was it added to satisfy a previous reviewer? It's not clear what you are saying here, or why the order in which you present results should be specified here.

Line 229 – do you mean "we first compared prevalence among sites...?"

Lines 249-250- it looks here as though you used the absolute mm² surface area covered by lesions as your response variable, but above you say you calculated the percent of surface area covered by lesions (i.e., you controlled for snake body size). Which variable did you use in this model? If the first, how can you account for different body size among individuals?

Lines 257-258 – how did you confirm that your burn-in length was sufficient?

Lines 264-265 - if this is the 558 snakes, move this sentence up. If not, please provide the sample size available for just these four species.

Lines 267-269 – Do you mean each mm² affected by lesions? Also, as above, this looks as though you used the surface area covered by lesions without controlling for snake size. Please clarify.

Line 270 – were multiple samples used for each individual snake in the analysis? If so, the rationale is unclear. If not, why did you use individual snake as a random effect? This is typically used to control for repeated measurements from an individual, but your study design doesn't appear to require this. (If I am simply misunderstanding, then consider this a nudge to revise so that everyone will understand what you mean.)

Did you test for an effect of snake size on detection of the fungus? Larger snakes may be more likely to carry detectable fungal loads even if they are not more likely to carry the fungus, simply because swabbing a larger snake up and down 5 times covers a larger area (likely to pick up more fungus) than swabbing a little snake. This is an important variable affecting detection probability and should be clearly addressed.

Results:

Lines 290-293 – this is where it will be helpful to provide more information about sampling dates. If you sampled Spain and Portugal later in the season, you would be less likely to detect the fungus. If not, that lends more strength to your inference of spatial variation in prevalence.

Line 395 – should be "positive"

Lines 398 – 402 – you may need/wish to rethink the "names" of Clades I and II, given that both clearly occur in Europe.

Figure 4: consider a more intuitive (and color-blind friendly?) color scheme. For example, you could use a color to denote samples containing two strains that is a mix of the colors used for those two strains. Try to find an option that will mesh well between the two panels. Currently, blue

and red mean different things in panels a and b.

Discussion:

The conclusions drawn here may be reasonable, but you need to clearly provide sampling date information so the readers can understand whether this variable may have affected the results.

Lines 470-471 – the sampling was uneven across the study area, and the borders of current countries are not biologically meaningful. Given the high spatial variation in prevalence in countries where the pathogen was detected, it seems premature to draw strong conclusions about which countries are “hotspots”, and the arbitrary political boundaries don’t help us understand the pathogen or disease. Can you reframe the discussion to focus on landscape features or environmental variation across the sampled area, that might explain the detected hotspots?

Line 469 – rogue parenthesis needs culling

Lines 473-474 – it is not. This is not accurate based on the current literature.

Lines 487-492 -this is a long sentence and is tough going. Revise?

Lines 495-495 – pathogen tolerance is another option here. If your working hypothesis is that the fungus was native to Europe and then introduced to NA (Ladner et al. 2022), it makes sense that snakes that co-evolved with the fungus might be tolerant to it, and develop lesions but not commonly experience mortality.

Lines 498 – 500 – What is missing from the discussion is a very clear acknowledgement that this study did not meet diagnostic criteria for ophidiomycosis. It did meet the standards used in many field studies (looking for lesions and testing for the fungus). But it did not meet diagnostic criteria for the disease, so it is possible 1) that the snakes that didn’t test positive at time of sampling did in fact have ophidiomycosis, but a biopsy would be required to confirm, or 2) that they had a different fungal infection (as you say). Important to distinguish between pathogen surveillance and clinical diagnosis.

Lines 512-532 – the content is good, but the paragraph is rambling. Can you revise to tighten this part up?

We thank the associate editor for their review of the comments and manuscript. We have addressed all of the components mentioned by the two reviewers and feel that the manuscript is greatly improved. We appreciate your consideration of the revised version.

Reviewer #1 (Remarks to the Author):

This manuscript describes a broad survey for ophidiomycosis in free-ranging snakes in Europe. The authors tested snakes for *Ophidiomyces ophidiicola* using qPCR of swabs, noted the presence and size of lesions, and conducted genotyping on the *O. ophidiicola* isolates. They found that geographic hotspots for *O. ophidiicola* existed in Switzerland, Ukraine, Germany and France, certain species of snakes were more likely to be infected with *O. ophidiicola*, and multiple clades of *O. ophidiicola* were present among sampled snakes, with *O. ophidiicola* prevalence at particular sites best predicted by snake species and pathogen clade.

Overall, this work represents a significant contribution to the understanding of ophidiomycosis in Europe. I commend the authors on addressing not only skin lesion and pathogen detection, but also measurement of clinical disease severity and genomic analysis. The manuscript is well-written with clear goals, methods, and results. The work could be reproduced based on the level of detail provided and the conclusions adequately supported by the findings. The figures are high-quality and well-explained.

Specific comments:

1. Be careful not to use the terms “disease,” “infection,” and “pathogen” interchangeably. Disease indicates the presence of clinical signs when the host responds to the presence of the pathogen, infection indicates the presence/growth of the pathogen in/on the host, and pathogen is the infectious agent. For example, on line 54 you refer to “disease hotspots of *Ophidiomyces ophidiicola*”, but *Oo* is the pathogen, not the disease. Is this paper addressing disease or prevalence hotspots?

Author’s response: Thank you for your comment, we agree and have rephrased accordingly “Here we elucidate the contribution of pathogen clade and host species in disease hotspots caused by *Ophidiomyces ophidiicola*, the pathogen responsible for snake fungal disease(...).” Line 54-55

2. Line 86: Suggest adding “the” between “theory suggests that” and “presence”

Author’s response: “the” was added in the sentence as suggested (line 87).

3. Lines 104-105: Suggesting adding a citation for Pribyl et al 2023 (*Oo* in snakes in the Czech and Slovak Republics, *Journal of Vertebrate Biology*, DOI: 10.25225/jvb.23050)

Author’s response: This citation was added as suggested (line 115).

4. Lines 210, 258: Are these the same genotypes and clades defined in the Ladner et al 2023 paper?

Author's response: Yes, although the Ladner paper performed whole genome sequencing, whereas in our paper, we classify *Oo* in clades based on ITS2 sequence only, so it is not as precise, and that is why we cannot differentiate clade II D/E as either clade IID or clade IIE in our paper (as explained on lines 386-389).

5. Figure 1a: “n” is missing in the sample size label for Spain (currently reads “Spain = 155”).

Author's response: Thank you for catching this, we have now corrected figure 1a.

6. Several Figures (1b, 2, 3): Very nice figures. The figure legends mention “black circles” for individual snakes testing positive or negative, but the circles appear gray in the figures.

Author's response: Thank you your comment, we have now corrected to “transparent black circles” (technically not grey but black with transparency setting) for all figure legends (lines 336, 357, 374, 376)

7. Figure S4: Photos are described as “mild”, “moderate” and “severe” in the figure legend, but these subjective terms have not been previously defined in the manuscript. Suggest leaving out the subjective descriptions.

Author’s response: We agree and have removed subjective wording as suggested.

8. Line 456: Consider rephrasing to “hotspots for *Ophidiomyces ophidiicola*” to more accurately reflect your results.

Author’s response: Thank you, we have added “*O. ophidiicola*” as suggested (line 432).

Reviewer #2 (Remarks to the Author):

This is an extremely interesting and timely study and I enjoyed reading it. I have concerns about the accuracy of the study framework and about some of the methods, but these would be simple to address in revisions. The study is worth polishing up – it is already impressive and will have greater impact with a few clarifications. I have made suggestions below, which I hope will be helpful.

General comments:

The likely impacts of ophidiomycosis are overstated, cherry-picking the literature to support an inaccurate “crisis” narrative. This is not helpful to our understanding of wildlife health, or to snake conservation. There is no need to overstate things: this disease is important to study even if it isn’t “the next chytrid or WNS”. Can you revise to present a more evidence-based perspective on the disease?

Authors’ response: We have revised our tone as suggested and have added literature to try to better reflect the current knowledge of the impact of this disease on snake populations (line 99-109).

Throughout the paper: consider using “ophidiomycosis” instead of “snake fungal disease”. SFD is an extremely vague name and there are a number of fungal pathogens that can cause disease in snakes. It is more correct (and recently, more common in the literature) to use the more specific term.

Consider not only susceptibility and resistance, but also tolerance to pathogens in your framework.

Authors’ response: We thank the reviewer for the suggestion of using “ophidiomycosis” instead of “snake fungal disease (SFD)”. After looking through Web of Science and associated citations we have decided to retain the use of SFD. This is primarily because we feel this name, while a little vague, appears more recognizable by a broader audience (SFD is referenced 3x more than

ophidiomycosis, even in the last year). Given the broad readership of *Communications Biology*, and the broader implications of our work beyond SFD we feel this would be most appropriate and accessible for a general biology audience.

Day of year for sample collection should be explicitly included as a factor in the analyses, in case the observed variation in prevalence is (partly or wholly) temporal, and not only spatial. Snake size (for example, using SVL) should be accounted for in analysis. Body size can affect detection probability for pathogens, because swabbing a larger surface area can pick up more fungal DNA.

Author's response: Thank you for your comment. We have now explored the relationship with the variables mentioned (i.e. date of capture and SVL), which have been added to the results. We chose to run these models separately instead of adding them all to the same model because adding more and more variables to try and understand multifactorial systems can provide inaccurate representation of each of these variables. Specifically, as you add more variables to a model, the precision of our predictions decreases because data contain a fixed amount of information. As we add more predictors, we spread the information in the data thinner and thinner. Therefore, the gain in accuracy from having more details (variables) in the model is outweighed by the loss of precision in estimating the effect of each variable. I have included a link that discusses these issues in more detail. <https://doi.org/10.32942/X2Z01P>

Somewhere, you should note that this study did not employ full diagnostic criteria for the disease. So, it speaks to detected pathogen presence and to observed lesions consistent with ophidiomycosis, but did not confirm clinical ophidiomycosis.

Author's response: We have added precision on line 191 “a snake is determined to be SFD positive if it was positive by qPCR (regardless of clinical signs being present or not).”

Lines 86-91: there is a third option here, supported by both theory and empirical evidence. If a species is tolerant to a pathogen (but not resistant), it can act as a reservoir host without experiencing population declines. This is how the fungus that causes WNS in bats persists in European bat populations. Some snakes in North America may also be tolerant to *Ophidiomyces*. For example, wild queensnakes do not exhibit reduced short-term survival (<https://esajournals.onlinelibrary.wiley.com/doi/abs/10.1002/eap.2251>) and wild eastern foxsnakes often recover from the disease during the active season (<https://doi.org/10.7589/JWD-D-21-00134>). Consider tolerance as an option, and revise other relevant pieces of the manuscript accordingly.

Author's response:

We have revised the text (line 88-91). This text was specifically referring to instances where we have high and low host susceptibility and low and high pathogen replication rates, which are in contrast and can result in hot and cold spots of prevalence. We are not addressing resistance or tolerance in this manuscript and had mentioned it only as one mechanism that can result in low host susceptibility. Given that there are multiple mechanisms that can drive this, we have reframed this section to account for different host mechanisms that can result in variable host susceptibility by referring to this more broadly. We appreciate the comment.

Lines 95-97: Ophidiomycosis was proposed as a serious threat to wild snakes in the papers cited, and others, but the aggregated data since then do not support this. Lorch et al (2016) is cited in Lines 99-100 without enough context. This study described a handful of population declines that were complex, with which the disease was associated but may not have been the main cause (as elegantly stated in that paper). You are not providing a balanced view of the data so far. In the previous line you cite Davy et al. (2021) to add to the list of affected snake species, but that study also summarized the evidence for impacts of ophidiomycosis on wild populations, and did not find support for the hypothesis that it is a serious threat. This could change in future as more populations are studied - but it seems appropriate to follow the available evidence for now.

Author's response: We agree and thank the reviewer for this comment. We have rephrased this entire paragraph, added context about the disease, as well as a more balanced description of the population-level impacts known to date by adding specific findings from the Davy et al (2021) review in Canada (lines 95-109). It is however important to note that the lack of support for an effect is not support for no effect.

Line 102 – delete “only”

Author's response: This word has been deleted (line 113).

Line 123 – the meaning of “we combined host species and pathogen clade” is unclear. It becomes more clear in the methods, but consider revising here to help the reader understand things earlier.

Author's response: We have rephrased this sentence to “Finally, we used an interactive model incorporating host species and pathogen clade to explore factors that may contribute to areas with high pathogen prevalence across the landscape” (line 133-134).

Methods:

This is impressive sampling and I applaud the team for coordinating this effort.

Author's response: Thank you!

Please clarify the following details, to ensure the methods could be replicated (or that variation among studies can be taken into account):

- How were bags and equipment sterilized between snakes?

Author's response: We have added description in the text “individual sterilized cloth bags or disposable paper bags, equipment was cleaned between snakes using 70% ethanol(...)”, see line 149-151.

- Were gloves worn to handle snakes? If so, were they changed between snakes? If not, how did you avoid cross-contamination of samples when handling snakes?

Author's response: Yes, disposable gloves were used between each snake, we added this in the text line 149.

- What was used to moisten the tip of the applicators?

Author's response: Sterile water was used to moisten the swab. We added more details to help clarify this on line 158.

- Prevalence of *Ophidiomyces* varies seasonally in North America. Please provide sampling dates for the sites, and consider incorporating ordinal date as a factor in your analyses.

Author's response: This is great point, and something we had looked at when we first started our analyses. We agree that this information is important, and we now have included a model investigating the effect of sampling dates on our detection data (see statistical methods, lines 244-248). Interestingly, and in contrast to what has been often observed in North America we found no support for an effect of date, which is in agreement with Haynes et al. (2020) that found no seasonal variation in the detection of *O. ophidiicola*. We have added a 2-panel figure in the supplemental material (Fig S2) showing the results of this model as well as a plot showing when each species was collected throughout the year.

- Lines 152-153 – can you confirm the meaning of “later”? Does this mean the cooler with ice and samples went directly to a lab and then a freezer, or was there different storage in between?

Author's response: Yes, the cooler with ice went directly to a lab and then to a freezer as soon as field work was over for that day. We have modified this sentence to “Swabs were individually stored in a 2 mL sterile tube in a cooler with ice while in the field and stored frozen at -20°C in the lab until analysis” (line 164)

- Line 178: “positive by qPCR” means the sample had a Ct value below some threshold, selected based on the standard curve. Can you specify the Ct threshold used? Was it

identical between the two labs? (N.B. – this is not intended to imply that results from the two labs were not comparable. It is simply important information to include regardless of what the details turn out to be.

Author’s response: That is correct, thank you for catching this. Yes, the same Ct threshold was used for both labs based on how our instruments performed using the same assay (our limit of detections were equivalent based on our standard curves). We have rephrased to clarify our method: “Samples that were positive were analyzed in duplicate, and a snake was determined to be SFD positive if any swab associated with that snake was positive by qPCR (regardless of clinical signs being present or not). Based on the limit of detection from the PCR assay, the Ct threshold was set at 39 for both labs.” Line 189-192.

This also applies to line 181- what criteria were used for “detection”? Was it different between the two labs?

The detection criteria were the same between the 2 labs (set by the Ct threshold now stated above). If any swab came back positive, the sample was analyzed for further sequencing (also confirming that *O. ophidiicola* was indeed present in that sample). We have rephrased: “Samples in which *O. ophidiicola* was detected by qPCR in at least one swab were subjected to follow up genotyping analysis” (line 195).

- Lines 165 and 173 describe two different types of negative controls that test for different potential sources of contamination (extraction blanks, and water). Were both negative controls run on the plates?

Author’s response: We apologize for the confusion. Extraction blanks were blanks used during the extraction step to make sure no contamination occurred during the extraction step (line 176-177). The water blanks (line 186) were blanks added to each PCR plate to confirm no contamination occurred during the PCR step. Yes, both negative controls (extraction and PCR) were run on each plate. We have added text to clarify in the lines mentioned above.

- Lines 221-223: it is not clear how you calculated surface area for snakes. Did you treat them as a cylinder? This wouldn’t be quite accurate, but it is justifiable if that’s what you did- it would give you a comparable SA per snake. Or did you do something different?

Author’s response: We apologize for the lack of information regarding how the surface area was calculated. Snakes were treated as a cylinder from the snout to the cloaca (knowing SVL and the diameter of the snake), and then as a cone from the cloaca to the tip of the tail (using the diameter at the base and tail length). We clarified in the text (line 236-237).

Line 144- should be “lesions”

Author's response: Thank you for catching this, it has been changed to “lesions” (line 155).

Line 157: could revise to “using the following methods”.

Author's response: This has been changed as “using the same method” (line 167-168).

Lines 215-223 – why did you not also consider the number of lesions as a measure of disease severity?

Author's response: We recognize that there are multiple ways to look at severity of infection. We wanted to quantify lesion surface area and control for the total size of the snake. We hypothesized that the greater total surface area covered in lesions would equate to higher disease severity. While the number of lesions and total surface area are likely correlated, we felt that surface area was a more accurate representation of what we were trying to estimate.

Lines 226-228: this text is odd. Was it added to satisfy a previous reviewer? It's not clear what you are saying here, or why the order in which you present results should be specified here.

Author's response: We have had this comment on previous reviews and agree that linking the methods with the results is both important but often challenging in the way we write scientific papers. We added it to help the reader link the analyses to the result section, and more easily be able to determine which variables and method was used for each analysis (also in Table S1).

Line 229 – do you mean “we first compared prevalence among sites...”?

Author's response: Yes, that is what we meant, and we have rephrased as suggested (line 249).

Lines 249-250- it looks here as though you used the absolute mm² surface area covered by lesions as your response variable, but above you say you calculated the percent of surface area covered by lesions (i.e., you controlled for snake body size). Which variable did you use in this model? If the first, how can you account for different body size among individuals?

Author's response: We have modified this analysis for clarity, and used the percentage of total surface area covered by lesions for each snake as our response variable, species as our predictor variable and site as our random effect. We used a beta distribution to fit this model and have updated the figure and table (results are unchanged). Line 257-260.

Lines 257-258 – how did you confirm that your burn-in length was sufficient?

Author's response: We used the diagnostic plots (which looked good) and we achieved model convergence, with all R-hat values below 1.01 (mentioned on line 300-301).

Lines 264-265 - if this is the 558 snakes, move this sentence up. If not, please provide the sample size available for just these four species.

Author's response: We have moved the sentence up for clarity (line 286-288).

Lines 267-269 – Do you mean each mm² affected by lesions? Also, as above, this looks as though you used the surface area covered by lesions without controlling for snake size. Please clarify.

Author's response: We did control for snake size in the analysis we previously used, however we have decided to modify this analysis for clarity. In the new model, we used the percentage of total surface area covered by lesions for each snake as our response variable, clade as our predictor variable and species as our random effect. We used a beta distribution to fit this model and have updated the supplemental figure and result section (results are unchanged). Line 290-292.

Line 270 – were multiple samples used for each individual snake in the analysis? If so, the rationale is unclear. If not, why did you use individual snake as a random effect? This is typically used to control for repeated measurements from an individual, but your study design doesn't appear to require this. (If I am simply misunderstanding, then consider this a nudge to revise so that everyone will understand what you mean.)

Author's response: We have addressed this comment in the response above.

Did you test for an effect of snake size on detection of the fungus? Larger snakes may be more likely to carry detectable fungal loads even if they are not more likely to carry the fungus, simply because swabbing a larger snake up and down 5 times covers a larger area (likely to pick up more fungus) than swabbing a little snake. This is an important variable affecting detection probability and should be clearly addressed.

Author's response: We thank the reviewer for asking this question. We have now added a model to test for the effect of snake size on the probability of Oo detection, but we restricted this analysis for sites that had Oo detected (using the prevalence estimated at each site). This allowed us to see if sites that were identified as hotspots in our paper (i.e. higher prevalence estimates) potentially had larger (>SVL) snakes sampled, which would confound our hypothesis that hotspots were driven by species and pathogen clade. We did not find support that sites with high prevalence had larger snakes and generally the distribution of snakes sampled was well mixed among sites within a species. We have now

included this model in our statistical methods (line 257-260) and the result section (line 324-326).

Results:

Lines 290-293 – this is where it will be helpful to provide more information about sampling dates. If you sampled Spain and Portugal later in the season, you would be less likely to detect the fungus. If not, that lends more strength to your inference of spatial variation in prevalence.

Author’s response: Thank you for your comment. We have added the results of the model investigating seasonality in our data (line 315-316). For reference, sampling was primarily targeted in the spring across all sites, no site or region was only sampled in a single season. We have added a 2-panel figure (Fig. S2) to show the capture dates of each individual snake. In panel (b) of Fig. S2, we graphed each snake’s capture event by date and species and can see that the species most collected in Portugal and Spain (*V. seoanei*, *V. aspis*, *V. latastei*) were collected over a wide range (and not just in the summer or fall).

Line 395 – should be “positive”

Author’s response: This has been corrected (line 380).

Lines 398 – 402 – you may need/wish to rethink the “names” of Clades I and II, given that both clearly occur in Europe.

Author’s response: We appreciate your comment. We plan to do whole genome sequencing in the future on the isolates we were able to collect from Europe, which will allow us to fully understand the phylogenetics of these strains. Until then, it is our preference to stay with the current nomenclature.

Figure 4: consider a more intuitive (and color-blind friendly?) color scheme. For example, you could use a color to denote samples containing two strains that is a mix of the colors used for those two strains. Try to find an option that will mesh well between the two panels. Currently, blue and red mean different things in panels a and b.

Author’s response: We have changed our color palette on Fig. 4a and 4b to address these concerns.

Discussion:

The conclusions drawn here may be reasonable, but you need to clearly provide sampling date information so the readers can understand whether this variable may have affected the results.

Author's response: We now have added a model looking for seasonality on SFD prevalence and found no significant variation in prevalence over our sampling period. We have also added a supplemental figure (Fig. 2S, panel b) that shows the sampling date for each individual snake, for each species. We hope this will address potential concerns.

Lines 470-471 – the sampling was uneven across the study area, and the borders of current countries are not biologically meaningful. Given the high spatial variation in prevalence in countries where the pathogen was detected, it seems premature to draw strong conclusions about which countries are “hotspots”, and the arbitrary political boundaries don't help us understand the pathogen or disease. Can you reframe the discussion to focus on landscape features or environmental variation across the sampled area, that might explain the detected hotspots?

Author's response: Thank you for your comment, we are in complete agreement. We have rephrased to frame the first paragraph of our discussion around site-level prevalence on line 437-440 (sampling unit defined in table S2, and throughout the manuscript). We included the countries in the text more for reference because we feel it helps the readers quickly orient themselves geographically. We have also added text to point out that environmental factors could also play a role in the hotspot pattern that we find (line 439-444). However, given the broad scale of our study, sites that appear to share similar environmental conditions sometimes had no SFD detection, while other were identified as hotspots, suggesting that host species and pathogen clade are more important factors. The role of environmental factors would have to be investigated more thoroughly and perhaps at smaller scales to really understand their importance, but this wasn't the scope of our study.

Line 469 – rogue parenthesis needs culling

Author's response: We have corrected this sentence (line 435-436).

Lines 473-474 – it is not. This is not accurate based on the current literature.

Author's response: This sentence has been revised and toned down (line 445-446).

Lines 487-492 -this is a long sentence and is tough going. Revise?

Author's response: This section has been revised and shortened (line 452-461).

Lines 495-495 – pathogen tolerance is another option here. If your working hypothesis is that the fungus was native to Europe and then introduced to NA (Ladner et al. 2022), it makes sense that snakes that co-evolved with the fungus might be tolerant to it, and develop lesions but not commonly experience mortality.

Author's response: Thank you for your comment. We have rephrased this sentence as: “The low prevalence and disease severity observed in Europe could be the result of lower pathogen virulence or decreased host susceptibility” which is more accurate than tolerance and resistance in this context (line 464-465).

Lines 498 – 500 – What is missing from the discussion is a very clear acknowledgement that this study did not meet diagnostic criteria for ophidiomycosis. It did meet the standards used in many field studies (looking for lesions and testing for the fungus). But it did not meet diagnostic criteria for the disease, so it is possible 1) that the snakes that didn't test positive at time of sampling did in fact have ophidiomycosis, but a biopsy would be required to confirm, or 2) that they had a different fungal infection (as you say). Important to distinguish between pathogen surveillance and clinical diagnosis.

Author's response: We have added text to this paragraph (line 470-473) to mention the possibility that qPCR negative snake (with lesions) could still have SFD, and that a tissue sample would be needed to confirm either way.

Lines 512-532 – the content is good, but the paragraph is rambling. Can you revise to tighten this part up?

Author's response: We have shortened this paragraph.

REVIEWERS' COMMENTS:

Reviewer #2 (Remarks to the Author):

Thank you for the thoughtful replies to my comments. The ms has been thoroughly and carefully revised, and I generally recommend publication.

I have only one minor concern. I appreciate that the authors will likely disagree, and that's fine, but I'm reiterating it here for consideration.

I do not understand why the SFD research community is so desperately attached to their crisis narrative. The revised introduction now reads: "Although population declines associated with SFD have been documented in some species of North American snakes (Lorch et al. 2016)..."  I have read Lorch et al. (2016), and it does not provide evidence for population declines caused by SFD.

In the rebuttal letter, the authors comment that "It is however important to note that the lack of support for an effect is not support for no effect." This is absolutely true. But when we have no evidence for an effect, the most parsimonious conclusion (*until other evidence arises*) is that there is no effect. As an absurd extension: There may be an invisible yeti in the corner of my room, and me not seeing it isn't evidence that it isn't there. Nevertheless, I think we can agree it is unlikely.

SFD is an important study system and this is a fabulous paper. It is not necessary to overstate the impacts of this disease to justify the research. The research stands on its own.

Thanks again for the thorough and excellent revision, and congratulations on an important study.

REVIEWERS' COMMENTS:

We once again thank the reviewer for careful reading of our manuscript and for the kind comments. We believe our paper is greatly improved after these 2 rounds of revisions.

Reviewer #2 (Remarks to the Author):

Thank you for the thoughtful replies to my comments. The ms has been thoroughly and carefully revised, and I generally recommend publication.

I have only one minor concern. I appreciate that the authors will likely disagree, and that's fine, but I'm reiterating it here for consideration.

I do not understand why the SFD research community is so desperately attached to their crisis narrative. The revised introduction now reads: "Although population declines associated with SFD have been documented in some species of North American snakes (Lorch et al. 2016)..."  I have read Lorch et al. (2016), and it does not provide evidence for population declines caused by SFD.

Author's response: We have revised this sentence to reflect the literature more accurately, and cited another paper instead of Lorch et al., 2016.

In the rebuttal letter, the authors comment that "It is however important to note that the lack of support for an effect is not support for no effect." This is absolutely true. But when we have no evidence for an effect, the most parsimonious conclusion (*until other evidence arises*) is that there is no effect. As an absurd extension: There may be an invisible yeti in the corner of my room, and me not seeing it isn't evidence that it isn't there. Nevertheless, I think we can agree it is unlikely.

SFD is an important study system and this is a fabulous paper. It is not necessary to overstate the impacts of this disease to justify the research. The research stands on its own.

Thanks again for the thorough and excellent revision, and congratulations on an important study.